# Synthesis of Cationic [4], [5], and [6]Azahelicenes with Extended π-Conjugated Systems

**Samuel Hrubý [1], Jan Ulč [1], Ivana Císařová [2] and Martin Kotora [1,***

[1] Department of Organic Chemistry, Faculty of Science, Charles University, Hlavova 8, 128 43 Prague, Czech Republic
[2] Department of Inorganic Chemistry, Faculty of Science, Charles University, Hlavova 8, 128 43 Prague, Czech Republic
* Correspondence: martin.kotora@natur.cuni.cz; Tel.: +420-221-951-058

**Abstract:** The scope of Rh-catalyzed C–C bond cleavage/annulation of biphenylene with various aromatic nitriles was studied. The subsequent Rh- and Ir-catalyzed C–H bond activation/annulation sequence of the formed 9-arylphenanthridines with alkynes gave rise to cationic [4], [5], [6] helical quinolizinium salts. The scope of the reaction with respect to the structural features of the starting 9-arylphenanthridines and alkynes was studied. Their helical arrangement was confirmed through single-crystal X-ray analyses of selected compounds. Most of the prepared quinolizinium salts exhibited fluorescence emission maxima in the region of 525–623 nm with absolute quantum yields up to 25%.

**Keywords:** C–C bond activation; C–H bond activation; helicenes; catalysis; rhodium; fluorescence





## 1. Introduction

The synthesis of purely carbon- and heteroatom-containing compounds possessing a helical scaffold has attracted considerable attention in the last several decades. Interest in these compounds is due to their potential applications in many fields of chemistry and related sciences [1–8]. In this respect, a number of synthetic strategies and approaches to access this class of compounds have been developed.

However, the synthesis of charged cationic helical compounds, e.g., cationic azahelicenes, has not been extensively studied, despite their potential applications. Nonetheless, such compounds have attracted attention, and as a result of that, a couple of synthetic strategies have been developed to access them. These routes have been based on: (a) photochemical cyclizations [9–11], (b) *N*-alkylation of the respective neutral azahelicene [12,13], (c) a three-component coupling reaction [14], (d) the catalytic cyclotrimerization of alkynylated azonia salts [15], I cross-coupling/C–H activation annulation sequence [16], and its enantioselective variant [17]. As far as applications are concerned, cationic helical compounds have been used as mitochondrial-targeted biomarkers [14], DNA intercalators [12], and *N*-doped nanographenes [16].

In view of the aforementioned, and also because of general interest to develop new procedures for the preparation of cationic aza-aromatic compounds, we were interested in developing a simple synthetic strategy that would allow preparing this class of compounds possessing laterally extended π-conjugated systems stretching beyond the basic helical scaffold, and to explore their photophysical properties.

In this context, we envisioned that such helical cationic azaromatic compounds could be prepared with a catalytic C–H bond activation/annulation sequence of the respective intermediates possessing the 2-phenylpyridine moiety (Scheme 1). The intermediates were expected to be synthesized either via catalytic C–C cleavage in biphenylene (**1**) followed by the insertion of nitriles forming 9-substituted phenanthridines and related compounds, or via a cross-coupling reaction between 9-chlorophenanthridine (**2**) with aromatic

organometals. As for the former, we have recently reported rhodium-complex-catalyzed C–C cleavage in biphenylene followed by reactions with various nitriles forming 9-phenyl or 9-pyridylphenanthridines [18]. We expected that such a process would constitute a simple and straightforward pathway to phenanthridine-based intermediates for the synthesis of angular and helical cationic aza-aromatics. In case the nitrile insertion reaction would not proceed as expected, the cross-coupling step would be used as an alternative.

**Scheme 1.** Original retrosynthetic analysis of helical quinolizinium salts with extended π-conjugated aromatic system.

Outlined here are our endeavors during the synthesis of novel cationic helical *N*-heterocycles by using catalytic procedures, the scope and limits of the individual reaction steps, as well as the photochemical (absorption and emission) and structural properties of the newly prepared cationic aza-aromatic compounds.

## 2. Results

### 2.1. Formation of 9-Substituted Phenanthridine Derivatives by C–C Bond Activation in Biphenylene and Insertion of Nitriles

Biphenylene (**1**) was synthesized according to the previously reported procedure in four steps [19]. Our previous results demonstrated that the nitrile insertion proceeded successfully with various benzonitriles and cyanopyridines by using two Rh-based catalytic systems ([Rh(COD)$_2$BF$_4$]/dppe or [Rh(COD)Cl]$_2$/dppe, microwave irradiation 190 °C) [18]. The former turned out to give better yields of the corresponding 9-aryl or 9-heteroarylphenanthridines; therefore, it was used for the subsequent C–C bond activation studies. Since our later results indicated that lowering the reaction temperature to 180 °C led to improved product yields, all C–C bond activation reactions were run by using [Rh(COD)$_2$BF$_4$] (10 mol%)/dppe (10 mol%) and microwave irradiation at 180 °C. Based on these facts, we initially studied the insertion reactions of three benzonitriles: benzonitrile (**3a**), 4-trifluorobenzonitrile (**3b**), and 4-methoxybenzonitrile (**3c**). In all cases, we obtained the corresponding products in generally good yields (Table 1). 6-Phenylphenanthridine (**4a**), 6-(4-methoxyphenyl)phenanthridine (**4b**), and 6-(4-(trifluoromethyl)phenyl)phenanthridine (**4c**) were obtained in 88, 47, and 85% isolated yields, respectively. Gratifyingly, the yield of **4c** was higher than the one reported by us previously [18] and demonstrated the beneficial effect of a lower reaction temperature.

**Table 1.** Reactions of **1** with benzonitriles **3a–3c**. (The reactions were run on 1 mmol scale with respect to **1**).

| Entry | 3 | R | 4 | Yield (%) [1] |
|-------|-----|------|-----|------|
| 1 | **3a** | H | **4a** | 88 |
| 2 | **3b** | OMe | **4b** | 47 |
| 3 | **3c** | CF$_3$ | **4c** | 85 |

[1] Isolated yields.

Then, we decided to explore the scope of the reaction with respect to nitriles bearing larger aromatic systems. For that purpose, we applied the above-mentioned conditions in reactions with 1-cyanonaphthalene (**3d**) (Table 2). Surprisingly, the desired product 6-(naphthalen-1-yl)phenanthridine (**4d**) was isolated in 3% yield only (Entry 1). In order to improve the yield of the reaction, catalytic systems using other ligands were tested as well. When dppp was used as a ligand (Entry 2), formation of product **4d** was not observed (according to a TLC analysis of the reaction mixture). The use of dppb (Entry 3) or PPh$_3$ (Entry 4) did not improve the reaction outcome either: **4d** was isolated in only 2% yields in both cases. The change from a Rh(I) catalyst for an Ir(I) one also did not lead to an improvement in the reaction yield and **4d** was isolated in only 1% yield (entry 5). According to the respective TLC analyses, the starting material remained unreacted and formation of side products in significant amounts was not detected.

**Table 2.** Reactions of **1** with benzonitriles **3d**. (The reactions were run on 1 mmol scale with respect to **1**).

| Entry | Catalyst | Ligand (mol%) | Yield (%) [1] |
|-------|----------|---------------|------|
| 1 | [Rh(COD)$_2$BF$_4$] | dppe (10) | 3 |
| 2 | [Rh(COD)$_2$BF$_4$] | dppp (10) | 0 |
| 3 | [Rh(COD)$_2$BF$_4$] | dppb (10) | 2 |
| 4 | [Rh(COD)$_2$BF$_4$] | PPh$_3$ (20) | 2 |
| 5 | [Ir(COD)Cl]$_2$ | dppe (10) | 1 |

[1] Isolated yields.

Since the insertion of 1-cyanonaphthalene (**3d**) into the C–C bond in biphenylene (**1**) did not proceed as expected, we attempted the insertion of 2-cyanonaphthalene (**3e**), which has a less sterically hindered nitrile group (Scheme 2). Running the reaction under the same reaction conditions as with **3e** provided the desired product 6-(naphthalen-2-yl)phenanthridine (**4e**) in 36% isolated yield. Carrying out the reaction at a higher temperature of 200 °C allowed obtaining product **3e** in 62% isolated yield. Its structure was unequivocally confirmed with single-crystal X-ray diffraction analysis (Figure 1a).

**Scheme 2.** Formation of phenanthridine **3e** by a reaction of biphenylene (**1**) with 2-cyanonaphthalene (**3e**). (The reaction was run on 01 mmol scale with respect to **1**).

**Figure 1.** ORTEP drawings of **4e** (**a**), **4f** (**b**), and **4g** (**c**). Ellipsoids are drawn with 50% probability.

A comparison of results obtained for the insertions of nitriles **3d** and **3e** indicate that the course of insertion of a nitrile after the C–C bond activation step is probably hampered by the steric hindrance of the nitrile group. Since the synthesis of larger helical cationic aza-aromatics would require the use of even more sterically hindered aromatic nitriles, we decided to change our strategy and to prepare the required phenanthridines by using cross-coupling routes.

*2.2. Synthesis of Phenanthridines Bearing 1-Napthyl, 9-Phenanthryl, and 4-Phenanthryl Moieties*

We decided to synthesize more structurally complex phenanthridines **4d**, **4f**, and **4g** via the Suzuki–Miyaura cross-coupling of 6-chlorophenanthridine **2** with the corresponding boronic acids **5a**–**5c**. 6-Chlorophenanthridine **2** was prepared from phenanthridin-6(5*H*)-one by dehydration–monochlorination with $POCl_3$ in 92% yield [20]. Boronic acids **5a** and **5b** were obtained from commercial sources and phenanthren-4-yl boronic acid **5c** was synthesized in four steps from 2-bromobenzaldehyde (for details, see comments in Supplementary Materials). The structure of **5c** was unequivocally confirmed with single-crystal X-ray diffraction analysis (Figure S4).

The cross-coupling reaction of **2** with **5a**–**5c** proceeded under standard conditions: $Pd(PPh_3)_4$ (0.2 mol%) as a catalyst and $K_2CO_3$ (6.3 eq.) as a base in a mixture of $THF:H_2O$ (5:1) and reflux [21] and provided the corresponding products in good isolated yields (Scheme 3). 6-(Napthalen-1-yl)phenanthridine (**4d**) was isolated in a nice 85% yield after 6 h of reaction time. The reaction of **2** with **5b** furnish 6-(phenanthren-9-yl)phenanthridine (**4f**) in 80% isolated yield. The cross-coupling with **5c** gave 6-(phenanthren-4-yl)phenanthridine (**4g**) in only 34% isolated yields. However, formation of other unidentified products was noted, resulting in a diminished yield of the desired product. The side products could not be separated from the unreacted starting material to determine their structures. The structures of **4f** and **4g** were unequivocally confirmed with single-crystal e. X-ray diffraction analyses (Figure 1b,c).

**Scheme 3.** Synthesis of **4d**, **4f**, and **4g** by Suzuki–Miyaura cross-coupling.

*2.3. C–H Activation/Annulation Sequence*

With a series of 6-arylphenanthridines in hand, we proceeded with the C–H activation/annulation sequence with various internal alkynes. Initially, we focused on the reactions of **4a**–**4c** with alkynes **6a**–**6c** by using a catalytic system comprising [Cp*RhCl$_2$]$_2$ (5 mol%), Cu(BF$_4$)$_2$ (1.5 eq.), and O$_2$ at 100 °C for 24 h that was shown to be suitable for the reactions of various arylated heterocycles with sp$^2$-*N* [22]. However, the isolated yields of the desired products were in the range of 0–44% only (Table S3). Further steps led to tuning the reaction conditions for a reaction of **4c** with **6c** (Table S4), which otherwise did not provide any product under the standard conditions. It turned out that it was necessary to increase not only the catalyst amount but also to change both the source of tetrafluoroborate anion as well as an oxidant and a solvent. Thus, a catalytic system composed of [Cp*RhCl$_2$]$_2$ (10 mol%), AgBF$_4$ (1 eq.), and Cu(OAc)$_2$ (1 eq.) at 100 °C for 24 h in DCE (Table S4) [23] gave the best results. Hence, further experiments were carried out under such conditions.

Thus, the reactions of 6-phenylphenanthridine (**4a**) with 1,2-diphenylethyne (**6a**) and 1,2-bis(4-methoxyphenyl)ethyne (**6b**) proceeded uneventfully, giving rise to 6,7-diphenylisoquinolino[2,1-*f*]phenanthridin-5-ium tetrafluoroborate (**7aa**) and 6,7-bis(4-methoxyphenyl)isoquinolino[2,1-*f*]phenanthridin-5-ium tetrafluoroborate (**7ab**) in 88 and 85% isolated yields, respectively (Table 3, Entries 1 and 2). Interestingly, a reaction with 1,2-bis(4-(trifluoromethyl)phenyl)ethyne (**6c**) took place as well; however, the expected product **7ac** turned out to be unstable and resisted full characterization.

**Table 3.** C–H activation/annulation of phenanthridines **4a**–**4c** alkynes **6a**–**6c**. (The reactions were run on 0.1 or 0.3 mmol scale with respect to **4a**–**4c**).

| Entry | 4 | R$^1$ | 6 | R | 7 | Yield (%) [1] |
|---|---|---|---|---|---|---|
| 1 | **4a** | H | **6a** | H | **7aa** | 88 |
| 2 | **4a** | H | **6b** | 4-MeOC$_6$H$_4$ | **7ab** | 85 |
| 3 | **4b** | OMe | **6a** | H | **7ba** | 85 |
| 4 | **4b** | OMe | **6b** | 4-MeOC$_6$H$_4$ | **7bb** | 90 |

[1] Isolated yields.

The reactions of 6-(4-methoxyphenyl)phenanthridine (**4b**) with **6a** and **6b** also proceeded well and furnished 9-methoxy-6,7-diphenylisochinolino[2,1-*f*]phenanthridin-5-ium tetrafluoroborate (**7ba**) and 9-methoxy-6,7-bis(4-methoxyphenyl)isochinolino[2,1-*f*]phenanthridin-5-ium tetrafluoroborate (**7bb**) in 85 and 90% isolated yields, respectively (Entries 3 and 4). As in the previous case, a reaction with **6c** provided an unstable product **7cc** that defied full characterization.

Surprisingly, all reactions of 6-(4-trifluoromethylphenyl)phenanthridine (**4c**) with the above-mentioned alkynes provided products **7ca–7cc** that were not stable and underwent uncontrollable degradation immediately after isolation (for details, see comments in Supplementary Materials). In this context, it is interesting that all compounds possessing 4-trifluoromethylphenyl moieties (**7ac**, **7bc**, and **7ca–7cc**) were not stable under ambient conditions. The only evidence for their formation is the presence of their respective molecular peaks in MS.

Then, we turned our attention to the reactions of **4d** and explored its reactivity in the C–H activation/annulation reaction with various alkynes. Initially, a reaction with diphenylethyne (**6a**) under the previously used conditions ([Cp*RhCl$_2$]$_2$ (10 mol%), AgBF$_4$ (1 eq.), and Cu(OAc)$_2$ (1 eq.) in DCE at 100 °C for 24 h) was tested (Table 4). The reaction furnished the respective product **7da** (6,7-diphenylbenzo[7,8]isoquinolino[2,1-*f*]phenanthridin-5-ium tetrafluoroborate) in a rather mediocre 27% yield (Entry 1). To increase the product yield, the amounts of additives were varied. In general, these attempts led to only a marginal improvement in yields (Table S5). Despite that, it turned out that using a catalytic system composed of [Cp*RhCl$_2$]$_2$ (10 mol%), AgBF$_4$ (1 eq.), Cu(OAc)$_2$ (1.1 eq.), and two equivalents of **6a** in DCE at 100 °C for 24 h gave the product in 29% isolated yield (Entry 2). Based on the above-mentioned results, the latter conditions were used for the subsequent reactions of **4d** with alkynes **6b–6c**. A reaction of with 1,2-bis(4-methoxyphenyl)ethyne (**6b**) led to the formation of **7db** (6,7-bis(4-methoxyphenyl)benzo[7,8]isoquinolino[2,1-*f*]phenanthridin-5-ium tetrafluoroborate), which was isolated in 31% yield (Entry 3). A reaction with 1,2-bis(4-(trifluoromethyl)phenyl)ethyne (**6c**) yielded **7dc** (6,7-bis(4-(trifluoromethyl)phenyl)benzo[7,8]isoquinolino[2,1-*f*]phenanthridin-5-ium tetrafluoroborate) in 21% isolated yield (Entry 4). Then, we decided to attempt a reaction with a representative of alkyl-substituted alkynes. For that purpose, we chose oct-4-yne (**6d**) and its reaction gave rise to **7dd** (6,7-dipropylbenzo[7,8]isoquinolino[2,1-*f*]phenanthridin-5-ium tetrafluoroborate) in 68% isolated yield (Entry 5). Perhaps, a smaller steric hindrance imparted by the *n*-propyl group in the annulation step can rationalize a higher reaction yield in comparison with that obtained in reactions with diarylethynes. The structure of **7da** was unequivocally confirmed with single-crystal X-ray diffraction analysis (Figure 2a).

**Table 4.** Formation of benzo[7,8]isoquinolino[2,1-*f*]phenanthridin-5-ium salts **7da–7dd**. (The reactions were run on 0.3 mmol scale with respect to **4d**.).

| Entry | Catalytic System [1] | R | 7 | Yield (%) [2] |
|---|---|---|---|---|
| 1 | *A* | Ph | **7da** | 27 |
| 2 | *B* | Ph | **7da** | 29 |
| 3 | *B* | 4-MeOC$_6$H$_4$ | **7db** | 31 |
| 4 | *B* | 4-CF$_3$C$_6$H$_4$ | **7dc** | 21 |
| 5 | *B* | *n*-Pr | **7dd** | 68 |

[1] Method A: [Cp*RhCl$_2$]$_2$ (10 mol%), AgBF$_4$ (1 eq.), and Cu(OAc)$_2$ (1 eq.) in DCE, 100 °C, 24 h; Method B: [Cp*RhCl$_2$]$_2$ (10 mol%), AgBF$_4$ (1 eq.), and Cu(OAc)$_2$ (1.1 eq.) in DCE, 100 °C, 24 h. [2] Isolated yields.

Next, phenanthridine **4e** was tested in a C–H bond activation/annulation sequence with alkynes **6a–6d** under the previously optimized reaction conditions (Scheme 4). The respective reactions proceeded in all cases, as can be judged by the disappearance of the starting material; however, the expected benzo[6,7]isoquinolino[2,1-*f*]phenanthridin-5-ium salts **7ea–7ed** turned out to be unstable and underwent continuous decomposition during column chromatography on silica gel (or with any other isolation techniquessuch as crystallization, etc.) and they could not be isolated as analytically pure substances.

Currently, it is not clear how and why the decomposition process takes place and what kind of products are formed. According to $^1$H NMR measurements, salts **7ea**–**7ed** decompose to a complex mixture of various substances. Therefore, the structures of the salts **7ea**–**7ed** were confirmed only through HRMS analyses.

**Figure 2.** ORTEP drawings of **7da** (**a**) and **7ga** (**b**). Ellipsoids are drawn with 50% probability.

**Scheme 4.** Attempts to obtain **7ea**–**7ed** through reactions of **4e** with alkynes **6a**–**6d**. (The reactions were run on 0.3 mmol scale with respect to **4e**.)

C–H bond activation 6-(phenanthren-9-yl)phenanthridine (**4f**) proved to be more challenging than in phenanthridines **4d** and **4e**. A reaction of **4f** with diphenylethyne (**6a**) under the previously optimized reaction conditions (method B, Table 4) did not proceed as expected and the respective product was obtained only in trace amounts (according to TLC analyses) and both starting materials were reisolated in almost quantitative amounts. Attempts to modify the composition of the catalytic system were not successful and the respective product was not formed in reasonable isolable amounts (Table S6).

Since the C–H bond activation in **4f** followed by annulation with **6a** was not successful, we attempted to carry out the reaction using a less sterically demanding alkyne, oct-4-yne (**6d**) (Table 5). A reaction under the optimized reaction conditions (method B, Table 4) provided the desired product **7fd** (18,19-dipropyldibenzo[5,6:7,8]isoquinolino[2,1-*f*]phenanthridin-17-ium tetrafluoroborate) in a rather low 10% isolated yield (Entry 1). Attempts to improve the yield by using [Cp*IrCl$_2$]$_2$ (10 mol%) as a catalyst (Entry 2) did not fare better and **7fd** was isolated in only 7% yield (Entry 2). When [Cp*Co(CO)I$_2$] (10 mol%) was used as a catalyst, the reaction did not proceed at all (Entry 3). The starting material was fully recovered from the reaction mixture.

**Table 5.** Screening of catalysts for C–H activation in **4f** and annulation with **6d**. (The reactions were run on 0.3 mmol scale with respect to **4f**.)

| Entry | Catalyst | Yield (%) [1] |
|-------|----------|-----------|
| 1 | [Cp*RhCl$_2$]$_2$ | 10 |
| 2 | [Cp*IrCl$_2$]$_2$ | 7 |
| 3 | [Cp*Co(CO)I$_2$] | 0 |

[1] Isolated yields.

Finally, 6-(phenanthren-4-yl)phenanthridine (**4g**) was subjected to a catalytic C–H bond activation/annulation reaction sequence with alkynes **6a**–**6d** (Table 6) under the previously optimized reaction conditions (method B, Table 4). First, a reaction with diphenylacetylene (**6a**) furnished **7ga** (6,7-diphenylnaphtho[2′,1′:7,8]isoquinolino[2,1-*f*]phenanthridin-5-ium tetrafluoroborate) in a reasonable 51% isolated yield (Entry 1). Its structure possessing the [6] helical scaffold was unequivocally confirmed with single-crystal X-ray analysis (Figure 2b). Subsequent reactions under the same conditions with alkynes **6b**–**6d** provided products **7gb** (6,7-bis(4-methoxyphenyl)naphtho[2′,1′:7,8]isoquinolino[2,1-*f*]phenanthridin-5-ium tetrafluoroborate), **7gc** (6,7-bis(4-(trifluoromethyl)phenyl)naphtho- [2′,1′:7,8]isoquinolino[2,1-*f*]phenanthridin-5-ium tetrafluoroborate), and **7gd** (6,7-dipropylnaphtho[2′,1′:7,8]-isoquinolino[2,1-*f*]phenanthridin-5-ium tetrafluoroborate) in isolated yields of 31, 21, and 20%, respectively (Entries 2–4).

**Table 6.** Formation of naphtho[2′,1′:7,8]isoquinolino[2,1-*f*]phenanthridin-5-ium salts **7ga**–**7gd**. (The reactions were run on 0.05–0.15 mmol scale with respect to **4g**).

| Entry | 6 | R | 7 | Yield (%) [1] |
|-------|-----|----------------------|-------|-----------|
| 1 | **6a** | Ph | **7ga** | 51 |
| 2 | **6b** | *p*-MeOC$_6$H$_4$ | **7gb** | 31 |
| 3 | **6c** | *p*-CF$_3$C$_6$H$_4$ | **7gc** | 21 |
| 4 | **6d** | *n*-Pr | **7gd** | 20 |

[1] Isolated yields.

### 2.4. Attempt of a Scholl Reaction of 7dd

The Scholl reaction is a method allowing to interconnect the carbon atoms of rotationally flexible aromatic compounds into more complex, rigid, π-conjugated planar systems. In this respect, we envisioned that compound **7dd** could be an interesting substrate to attempt its conversion into a planar substance with condensed aromatic rings. The Scholl reaction is commonly carried out by using strong Lewis acids, but other less drastic methods are available as well [24,25]. To avoid higher temperatures and strongly Lewis-acidic reagents, a previously developed procedure based on the use of DDQ (8 eq.), TFA (315 eq.) in DCM at 0 °C for 30 min under an argon atmosphere was used to intramolecularly cyclize **7dd** [26]. Although **7dd** was totally consumed under these conditions, a complex reaction

mixture was obtained. After considerable effort, a fraction was isolated—the MS analysis of which showed a molecular peak that could be attributed to the elusive product **8** (14,15-dipropylnaphtho[2′,1′,8′:4,5,6]quinolino[1,8,7-*fgh*]phenanthridin-13-ium tetrafluoroborate). Attempts to record high-resolution $^1$H or $^{13}$C NMR were not met with success due to the low solubility of the isolated substance; hence, the recorded MS data are the only evidence supporting its formation.

### 2.5. Structural Properties

Suitable crystals for single-crystal X-ray diffraction analysis were obtained for boronic acid **5c** (Figure S4), phenanthridines **4e**, **4f**, and **4g** (Figure 1a–c), and for cationic helical aza-aromatic compounds **7da** and **7ga** having the [5]- and [6] helical scaffolds, respectively (Figure 2a,b).

In case of **7da**, the sum of three dihedral angles derived from five C–C bonds, which should reflect the degree of molecular twist, was 71.7°. The degree of molecular twist is bigger than the one for parental all-carbon [5] helicene (66.4°) [27]. The sum of the of four dihedral angles derived from six C–C bonds in **7ga** was 93.6°, which is bigger than the one in parental all-carbon [6] helicene (87.5°) [28]. The higher helical pitch could be a result of steric hindrance exerted by the presence of the benzo moiety attached to the second aromatic ring.

### 2.6. Photochemical Properties

Out of a library of 6,7-bis(aryl)isoquinolino[2,1-*f*]phenanthridin-5-ium tetrafluoroborates **7aa**–**7cc**, absorption and emission spectra were recorded for selected representatives as dichloromethane solutions (see Chart S1 in Supplementary Materials). The emission fluorescence maxima were in the range of 521–650 nm with rather low quantum yields ($\Phi_{abs}$) in the range of 8–99% (Table 7).

**Table 7.** Emission maxima and quantum yield of compounds **7aa**, **7ab**, and **7ba**–**7bc**.

| 7 | $\lambda_{ems}$ (nm) [1] | $\Phi_{abs}$ (%) [1] |
|---|---|---|
| **7aa** | 540 | 25 |
| **7ab** | 650 | 8 |
| **7ba** | 521 | 22 |
| **7bb** | 622 | 19 |
| **7bc** | 492 | >99 |

[1] Measured in DCM solutions ($c = 10^{-6}$ M).

The absorption and emission spectra of benzo[7,8]isoquinolino[2,1-*f*]phenanthridin-5-ium salts **7da**–**7dd** and naphtho[2′,1′:7,8]isoquinolino[2,1-*f*]phenanthridin-5-ium salts **7ga**–**7gd** were recorded as dichloromethane solutions (see Chart S2 in Supplementary Materials). The emission fluorescence maxima were in the range of 525–623 nm and the quantum yields ($\Phi_{abs}$) were in the range of 7–20%. Compounds **7dd** and **7gd** possessing the *n*-Pr groups did not exhibit any fluorescence at all (Table 8).

**Table 8.** Emission maxima and quantum yield of compounds **7da**–**7dd** and **7ga**–**7gd**.

| 7 | $\lambda_{ems}$ (nm) [1] | $\Phi_{abs}$ (%) [1] |
|---|---|---|
| **7da** | 537 | 6 |
| **7db** | 623 | 14 |
| **7dc** | 525 | 6 |
| **7dd** | – | 0 |
| **7ga** | 575 | 8 |
| **7gb** | 616 | 20 |
| **7gc** | 578 | 7 |
| **7gd** | – | 0 |

[1] Measured in DCM solutions ($c = 10^{-6}$ M).

## 3. Materials and Methods

All experimental procedures, compound characterization, X-ray diffraction data, and copies of [1]H and [13]C spectra are available in the Supporting Information.

## 4. Conclusions

The synthesis of 9-arylphenanthridines using a catalytic C–C bond cleavage/nitrile insertion sequence was successful only in the cases of **4a**–**4c** and **4e**. In the case of other substrates, the yields of products were very low (**3d**) or the reaction did not proceed at all (**4f**). It seems that the course of the reaction is sensitive to steric hindrance in both reactants. The required 9-arylphenanthridines (**4d**, **4f**, and **4g**) were successfully prepared through a Suzuki–Miyaura cross-coupling reaction of 6-chlorophenanthridine with the corresponding arylboronic acids. In general, the catalytic C–H bond activation/alkyne annulation sequence proceeded with all 9-arylphenanthridines, giving rise to the corresponding cationic 6,7-bisarylisoquinolino[2,1-*f*]phenanthridin-5-ium salts **7aa**–**7cc** with the [4] helical scaffold in good yields. Structurally related benzo[6,7]isoquinolino[2,1-*f*]phenanthridin-5-ium salts **7ea**–**7ed,** also possessing the [4] helical scaffold, were synthesized in higher yields, but were unstable, underwent decomposition, and could not be isolated in an analytically pure form. In the case of benzo[7,8]isoquinolino[2,1-*f*]phenanthridin-5-ium salts **7da**–**7dd** possessing the [5] helical scaffold, these were prepared in average yields. Phenanthridine **3f** turned out to be an unsuitable substrate for the catalytic C–H bond activation/annulation sequence and it did not react with diarylethynes. It proceeded with oct-4-yne (**6d**) only and furnished the corresponding product **7cd** in low isolated yield (10%). A higher reactivity was also observed in the case of **4g**, the reactions of which yielded naphtho[2′,1′:7,8]isoquinolino[2,1-*f*]phenanthridin-5-ium salts **7ga**–**7gd** with the [6] helical scaffold in good yields. Obviously, steric hindrance plays an important role in this process as well. Emission spectra were recorded for compounds **7da**–**7dd** and **7ga**–**7gd** that had fluorescence emission in the region of 525–623 nm with absolute quantum yields in the region of 0–20%. Single-crystal X-ray diffraction analyses of the phenanthridines **4b**, **4c**, and **4f** and cationic salts **7aa** and **7da** confirmed their structure.

**Supplementary Materials:** Experimental procedures, spectra characteristics, and copies of [1]H and [13]C NMR spectra can be downloaded at: https://www.mdpi.com/article/10.3390/catal13050912/s1. Chart S1: Absorption spectra of compounds **7aa**, **7ab**, and **7ba**–**7bc** measured in DCM solutions ($c$ = 10[−5] M) (**1a**). Normalized emission spectra of compounds **7aa**, **7ab**, and **7ba**–**7bc** (**1b**); Chart S2: Absorption spectra of compounds **7da**–**7dd** and **7ga**–**7gd** measured in DCM solutions ($c$ = 10[−5] M) (**1a**). Normalized emission spectra of compounds **7da**–**7dc** and **7ga**–**7gc** (**1b**); Figure S1: ORTEP drawing of **7da**. Ellipsoids are drawn with 50% probability; Figure S2: ORTEP drawing of **4f**. Ellipsoids are drawn with 50% probability; Figure S3: ORTEP drawing of **4e**. Ellipsoids are drawn with 50% probability; Figure S4. ORTEP drawing of **5c**. Ellipsoids are drawn with 50% probability; Figure S5: ORTEP drawing of **4g**. Ellipsoids are drawn with 50% probability; Figure S6: ORTEP drawing of **7ga**. Ellipsoids are drawn with 50% probability; Table S1: Suzuki-Miyaura cross-coupling of 2-bromobenzaldehyde with 2-bromophenylboronic acid under various conditions; Table S2: Screening of the Suzuki-Miyaura cross-coupling reaction conditions; Table S3: Screening of Rh-catalyzed C–H activation/annulation sequence of phenanthridines **4a**–**4c** with alkynes **6a**–**6c**; Table S4: Screening of various reaction conditions for C–H activation/annulation of **4c** with **6c**; Table S5: Screening of additives' amounts in the C–H bond activation/annulation reaction of **4d** with **6a**; Table S6: Screening of conditions for Rh-catalysed C–H activation/annulation sequence in phenanthridine **4f** with diphenylethyne (**6a**); Table S7: Crystal data, data collection, and refinement parameters for **7da**, **3f**, **3e**, **5c**, **3g**, and **7ga**. References [21–23,26,29–46] were cited in Supplementary Materials.

**Author Contributions:** Conceptualization, M.K.; formal analysis, M.K., J.U. and S.H.; experimental work, J.U. and S.H.; X-ray, I.C.; writing—original draft preparation, M.K.; writing—review and editing, M.K., J.U. and S.H.; supervision, M.K.; project administration, M.K.; funding acquisition, M.K. and J.U. All authors have read and agreed to the published version of the manuscript.

**Funding:** The authors gratefully acknowledge the financial support from the Czech Science Foundation (grant no. 21-29124S) and the Charles University Grant Agency (grant no. 1190218).

**Data Availability Statement:** The data presented in this study are available in the Supporting Information section.

**Acknowledgments:** The data used in the text were obtained by S.H. and J.U. during the implementation of their bachelor's, diploma, and PhD projects.

**Conflicts of Interest:** The authors declare no conflict of interest.

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
