# Peer review of "Synthesis of Cationic [4], [5], and [6]Azahelicenes with Extended π-Conjugated Systems"

_catalysts, doi:10.3390/catal13050912_

Round 1

Reviewer 1 Report

This paper describes the synthesis of cationic azahelicenes via Rh- and Ir-catalyzed C-H activation/annelation reactions of 9-arylphenanthridines with alkynes. This paper deserves to be published in Catalysis because it provides a new method for the synthesis of cationic azahelicenes, which have been synthesized only rarely so far. However, the following points need to be corrected.

 1) There are many errors in compound numbers, Tables, and Scheme captions. As far as I have found, they are as follows.

1. page 3, line 100: 2a-2c 2d

2. page 3, line 104: 2a 2d

3. page 3, line 107: 2d 2e

4. page 4, line 141: It is not specified which boronic acid 5a-5c in Scheme 3 refers to.

5. page 5, line 150: "C, Table S4" is sufficient for "Table S4".

6. page 5, line 158: Entries 1-3 Table 3, Entries 1-3

7. page 5, line174: alkynes 5a-5c and alkynes 6a−6c.

8. page 6, line 200: 7 7da-7dd

9. page 7, line 240: 6fd  7fd

10. page 7, line 242: 7cd 7fd

11. page 8, line 260: 3c 3f

12. page 8, line 276: 7da-7dd 7ga-7gd

13. page 8, line 280: 7fd 7dd

14. page 8 line 283: 7fd 7dd

15. page 9, line 288: 7fd 7dd

16. page 9, line 289: 7fg 7dd

17. page 9, line 297: 7fd 7dd

 2) page 9, line 301-303: This sentence, ”In case of 7da and 7ga the sums of the of 3 or 4 dihedral angles derived from five or six C–C bonds, which should reflect the degree of molecular twist, were 71.7 and 93.6°, respectively.” does not make sense. It should be more specific.

 3) The reviewer thinks the section 2-4 (attempt of a Scholl reaction) is unnecessary for this paper.

 4) The vertical axis of absorption spectrum should be absorption coefficient instead of absorbance (if possible). In the emission spectrum, the vertical axis should be written as “Normalized Intensity”, not “Intensity/a.u.”.

Reviewer 2 Report

This manuscript written by Kotora et al. describes synthetic studies of helical quinolizinium salts from 9-arylphenanthridines through rhodium-catalyzed C–H bond activation followed by cycloaddition with alkynes. The 9-arylphenanthridines were prepared by C–C bond cleaving cycloaddition reaction of biphenylene with arylnitriles in the presence of rhodium catalyst, whose catalytic system has been developed by the same authors in the previous work. Although the reaction with 1-naphthonitrile poorly proceeded, the target molecule could be obtained through another route utilizing palladium-catalyzed cross-coupling reaction of 9-chlorophenanthridine with 1-naphthylboronic acid in good yield. The palladium catalyst also allowed access to 9-phenanthrenylphenanthridines. The prepared 9-arylphenanthridines were subjected to rhodium-catalyzed cycloaddition with alkynes to afford helical quinolizinium salts. The helical structures of the products were unambiguously determined by X-ray crystallographic analysis. This work can attract synthetic interest and have enough impact for publication in Catalysis. However, there are some points to be improved and some questions from the reviewer. The following comments will be beneficial for improvement of the quality of the manuscript.

1) In Table 1 and the corresponding main text, the authors note that yield of 3a is 88%, which is higher than that reported by the same authors [ref. 18]. However, the reported yield of 3a in ref. 18 is 92%. There is inconsistency.

2) Some quinolizinium salts (7ac, 7bc, 7ca, 7cb, 7cc, 7ea, 7eb, 7ec, and 7ed) products are reported as unstable and immediately undergo decomposition. Nonetheless, isolated yields of these compounds are described. Are these values valid? If so, the authors should provide spectral data to supports the purity of these compounds in the isolated form. Validity of the value of isolated yield should be supported by some experimental data.

3) Scholl reaction product 8 was not analyzed by NMR due to its low solubility. To support the isolation and purity of the compound 8, elemental analysis should be attempted.

4) Emission spectrum of 7ab has larger noise compared to that of other compounds. Does this spectrum have enough reliability?

5) Datum of 7bc is missing in Table 8. Emission maxima of 7bb looks over 600 nm from Chart 1b, albeit the maxima in Table 8 is 402 nm. Is it true?

Reviewer 3 Report

In the article titled ""Synthesis of cationic ...........pi-conjugated systems", Ruby et al reported a method of synthesis of cationic [4], [5], and [6]Azahelicenes. The work is nicely described. It is a well written article. The present work and the work described in the reference 16 discuss the same methodology  except that the methodology is applied to slightly different starting materials. Also, many products do not have complete characterization data. Mass data alone will not assure the purity of the compound. The article can be reconsidered after a major revision addressing the following concerns.

In the fourth paragraph, first page explaining the Scheme 1, biphenyene is numbered 1 and In the scheme 1 biphenylene is not numbered.

For the compound [Rh(COD)2BF4], COD is written in small letters in the text part and in capital letters in scheme. Generality needs to be maintained.

In all the Tables, experimental details along with he scale of the reaction need to be included.

In Table 2, the isolated yields are given in the values which is lower than the experimental errors.

For all the new compounds, full characterization data including 1H NMR spectra, 13C NMR spectra and mass data need to be provided. Vey important products (7ea - 7ed) highlighting the significance of the methodology were not characterized by NMR spectroscopy. It is mentioned that the product got decomposed during the chromatography and pre compounds were not isolated. But, it the Table 5, isolated yields were reported. Need more explanation on this. It is necessary to optimize the purification strategy to get the produces for NMR characterization.

The section 'Attempt of a Scholl reaction of 7fd' can be removed as there is no clear evidence on the product formed. The product is characterized only by HRMS. The yield of the reaction is very low and NMR characterization is not provided.

In many synthetic steps, yields were very low (< 10%) and the scale of the reactions is not provide. Hence it very hard to know the amount of product obtained and reproducibility of the data.

It is very hard to understand the rationale behind the presence of absorption and emission data, in the manuscript. The authors did not explain the photochemical properties instead simply provided the data.

In the SI section, the optimization studies (Page number: 6) need to be brief and completely rewritten. 

Round 2

Reviewer 3 Report

It is great that the authors have addressed several issues in the revised manuscript. But, there are some issues to be addressed in the revised manuscript. The article can be accepted after resolving the following issues.

Concern in numbering sequence: In the text, compound 4 is appearing before compound 2 and 3. It is the same with scheme. Uniform numbering is expected in the article. Even in Table 1, numbering is non-sequence (entries 1, 2 and 3 are 2b, 2a and 2c respectively. Authors can rearrange them to be in  a sequence as 2a, 2b and 2c. 

The general practice in optimizing a reaction condition is to perform the reaction for a pair of substrate (For example 3c and 6c) under different reaction condition to find out the optimized condition and use that condition for the other combination of substrates. In this manuscript, it is carried out other way (Table S3 and followed by Table S4) which is not acceptable. Also, in table S4, the amount of catalyst keeps varying for the different entires, leading to lot of confusion to the readers. Hence, the content in Table S3 need to be removed and the related text can be rephrased. Also, Ekv.  In Table S4 Is not a general abbreviation.

For several of the compound (7ac, 7bc, 7ca, 7cb, 7cc, 7ea, 7eb, 7ec, 7ed, 8), authors have mentioned that the compounds were isolated and immediately after isolation they got decomposed, hence characterized only by mass techniques. The mass technique alone can not be considered as an evidence for the synthesis of compound. The author should remove the corresponding entries rather than claiming the synthesis of the compound. In the case of claiming the synthesis, the author should provide the complete characterization data.

Line # 192 states that alkynes 6b-6d were used. It is raising three questions, while following the manuscript. (1) It is very hard to find the structural detail of 6d. (2) All of a sudden why the authors are using the compound 6d? (3) why the compounds 3a-3c are not reacted with the alkyne 6d?

Table S4 has the heading Catalytic system (A, B and C) and the following paragraph explains Procedure (A, B and C). Are they both the same? Need uniformity in this context.

The paragraph following Table S5 explains ‘Procedure D’. Where is the ‘Procedure D’ is used?

In the Table 3, compounds derived from 3c and 6a/6b/6c are having stability issue. What about the stability of compound derived from 3c and 6d. Is it stable enough to get a neat NMR spectrum?

SI section Page# 30, all of a sudden, the section 3.4 discusses about the reaction of 3c with 6a. Are the compound numbers in this paragraph correct?

Product in Table S6, is it 7fa or 7ca?

In line # 290, the authors state that the compound 7dd could be an interesting substrate for Scholl reaction. Authors failed to confirm the formation of Scholl product from this substrate 7dd, they should try with other substrates (7da-7dc) having a similar structure to highlight the future endeavors in this area.

The emission spectra are too much noisy, not of publication quality.

In SI section, the compounds 1,2-Diethynylbenzene and 2,3-Bis(trimethylsilyl)biphenylene are numbered 85 and 85 respectively. What is the logic behind this numbering?

In the statement appearing in the synthesis of 1,2-diethynylbenzene, “Bis((trimethylsilyl)ethynyl)benzene (38.8 mmol, 10.5 g) was dissolved in a mixture of methanol and chloroform (2×20 mL)”, why is the amount of solvent is given as 2X20 mL instead of 40 mL?

Typo error in the compound names: 

(1) Bis((trimethylsilyl)ethynyl)benzene => 1,2-Bis((trimethylsilyl)ethynyl)benzene (in all instances it is appearing)

(2) phenantridine => phenanthridine (in several instances)

(3) fenyl => phenyl (in two instances, 7ca na d7cb name)

Author Response

See the attached file: reviewer-3-replies
